# The Advantage of Case-Tailored Information Metrics for the Development of Predictive Models, Calculated Profit in Credit Scoring

**DOI:** 10.3390/e24091218

**Published:** 2022-08-30

**Authors:** Daniel Chrościcki, Marcin Chlebus

**Affiliations:** Faculty of Economic Sciences, University of Warsaw, 00-241 Warsaw, Poland

**Keywords:** credit scoring, econometrics, machine learning, performance metrics, model development, EMP, CP

## Abstract

This paper compares model development strategies based on different performance metrics. The study was conducted in the area of credit risk modeling with the usage of diverse metrics, including general-purpose Area Under the ROC curve (AUC), problem-dedicated Expected Maximum Profit (EMP) and the novel case-tailored Calculated Profit (CP). The metrics were used to optimize competitive credit risk scoring models based on two predictive algorithms that are widely used in the financial industry: Logistic Regression and extreme gradient boosting machine (XGBoost). A dataset provided by the American Fannie Mae agency was utilized to conduct the study. In addition to the baseline study, the paper also includes a stability analysis. In each case examined the proposed CP metric that allowed us to achieve the most profitable loan portfolio.

## 1. Introduction

Predictive model development is a multifaceted task involving steps, such as preparing a dataset, selecting an appropriate algorithm, optimizing hyperparameters and determining the optimal cut-off point. While the described stages usually receive considerable attention [1,2,3], an often overlooked issue is the selection of the information metric against which the model is optimized. This issue turns out to be no less important than the other steps, especially if we are dealing with an imbalanced dataset or a case in which the misclassification error costs are significantly different for a false positive and a false negative case [4]. The most commonly used metrics, such as Area Under the ROC curve and the Kolmogorov–Smirnov test [5], unfortunately do not account for this disparity and are not best suited for credit scoring [6].

The problem of misclassification costs is not completely ignored in credit scoring. Both in the literature and real business applications, there are proposals for metrics that take this phenomenon into account [6,7]. However, most of the model development methods incorporating this problem focus only at the final stage of selecting a cut-off point such that the proportion of errors is inversely proportion to their costs [8,9]. At the same time, the usage of a problem-suited metric for the whole model tunning process may allow obtaining a model better suited to the problem [10].

The overarching goal of building predictive models for credit scoring is to increase a credit institution’s profitability by differentiating profitable (often referred to as good) and lossy (bad) customers. In this context, the primary purpose of this work is to demonstrate that synthetic metrics, commonly used for credit risk models’ development, can lead to suboptimal solutions in business applications. The paper aims to show that using a problem-tailored metric, Calculated Profit, developed within this research, for the whole model development process, allows one to achieve a significant advantage in terms of overall credit portfolio profitability.

The study is organized as follows: Section 2 is devoted to the issue of classification models errors and methods for evaluating their performances. It also presents the CP metric proposed in the paper for the optimization of credit risk models. Section 3 introduces the methodology and results of the main study, including a description of the utilized dataset. Section 4 is dedicated to a sensitivity analysis of the compared metric. The conclusions and possible further development directions are presented in Section 5.

## 2. Probability of Defaulting

### 2.1. Credit Risk Modeling

The ability to distinguish between a potentially good and bad customer is of tremendous value to a financial institution, dramatically impacting corporate profits and the riskiness of a loan portfolio. Ideally, it would be possible to lend to all profitable customers while rejecting applications of the risky and potentially lossy ones. To some extent, such insight can be provided by developing a credit risk model. The task of such a model in the field of information theory is to reconstruct information about the riskiness of a loan using signals, which are random variables (features) describing the characteristics of that loan. These signals are usually significantly distorted, so the role of a good model is to extract only those pieces of the information that allow for a correct assignment of riskiness for a given set of loan features. The results of such a model are then used to decide whether or not to grant credit to a particular customer. Considerable attention has been given to this issue of risk modeling [11,12,13] and a broad spectrum of econometric and machine learning methods have been used for the described task [14,15].

### 2.2. Model Performance Metrics

Credit risk modeling is usually done by determining the probability of defaulting for a given customer in the range from 0 to 1. Then, based on a settled or determined value, usually called a threshold or cut-off point, customers with a risk below that value are granted credit, whereas those with a probability above it are rejected. As a whole, this process is referred to as binary classification, meaning labeling customers as good (“0”) or bad (“1”). When dealing with this issue, four model results are possible in terms of the prediction correctness. They are customarily presented by means of a contingency table (Table 1):

Two of the results: true positives and true negatives, stand for correct identifications of bad and good credits, respectively. The two following outcomes of false positives and false negatives, marked in gray in Table 1, represent incorrect classifications, which in the business application of the model, are usually associated with specific costs for the company. Moreover, those costs are usually significantly different for those two types of errors [16]. Relative to an ideal discriminator, the model payoff in a credit scoring application can be represented as (Table 2):

In general, the goal of model building is to achieve the lowest possible prediction error. There are many widely used metrics for evaluating models [17], including, for example:(1)$$Sensitivity=\frac{TP}{TP+FN}$$
(2)$$Specificity=\frac{TN}{TN+FP}$$
(3)$$Precision=\frac{TP}{TP+FP}$$
(4)$$Accuracy=\frac{TP+TN}{TP+TN+FP+FN}$$

Although the above measures are widely and readily used, they have numerous drawbacks that eliminate them from application in the complex development of advanced models. First of all, they are not suitable for imbalanced datasets [18]. Secondly, they depend on the choice of a cut-off point, which is used for transferring from the probability plane, returned by the predictor, to the binary classification. This fact is problematic, as this point is usually selected only at the last stage of model development. For those reasons, more universal metrics, such as Area Under The ROC Curve (AUC) (Figure 1) [19], which measures discriminative power of a model through the whole spectrum of possible cut-off points and is partially insensitive to the class imbalance problem, are most commonly used for the model development process [20].

Similar characteristics are offered by the Kolmogorov–Smirnov test (KS-statistic) [21] or Area Under Precision–Recall Curve (AUPRC) [22]. However, all these measures are still not ideal for evaluating models used in business applications because they do not consider misclassification costs, assuming that they are equal for FP and FN prediction. In the example of crediting institutions, if the bankruptcy cost exceeds the profit on good credit, out of two models with identical accuracy or AUC, the one that achieves a smaller percentage of false negative classifications is preferred.

### 2.3. Cost-Sensitive Methods

Many attempts were made to account for the cost aspect [6,7,8,23,24]. One of the simplest methods uses a graph of the Receiver Operating Characteristic (ROC) (Figure 2) [25]. In this approach, the optimal cut-off is determined as a point on the ROC curve, for which it is tangent to a line with a slope coefficient equal to the quotient of the false negative and false positive costs:(5)$$f^{\prime }\left(misclassification\;costs\right)=\frac{Cost(False\;Negative)}{Cost(False\;Positive)}$$

This approach should theoretically ensure that the proportion of each type of false prediction is inversely proportional to their cost in a loan portfolio. However, the presented method allows for including misclassification costs only in the final step of selecting the cut-off point. The whole training process of the model in this case is performed solely using AUC or another similar metric. Thus, misclassification costs are not fully taken into account, and it has been proven that an additional advantage can be gained by using metrics considering misclassification costs throughout the whole optimization stage [26]. Some authors already addressed this issue in the area of credit scoring [27,28]. There were also proposals for new metrics incorporating cost aspects. Hand [6] presented an $M_{2}$ measure. Calabrese [8] introduced the MEL curve. One of the newest and most sophisticated ones, which we will focus on in this study, is the Estimated Maximum Profit (EMP) metric, described by Verbraken [7].

The main idea behind EMP is to estimate the costs of false positive and negative predictions as proportions of the Initial Principal Balance (IPB), namely, the size of the loan, calculated as follows:(6)Cost(FN)=E(LGD)
(7)$$Cost\left(FP\right)=ROI=\frac{I}{IPB}$$
where I = total interests, LGD = Loss Given Default as a proportion of the IPB and ROI = Return on Investment.

The Loss Given Default (LGD) distribution is assumed to come from a point-continuous distribution, defined as below:(8)$$f_{LGD}=\left\{\begin{array}{c} p_{0},\;for\;LGD=0\\ 1-p_{0}-p_{1}\;for\;LGD\;\epsilon \;\left(0,1\right)\\ p_{1}\;for\;LGD=1 \end{array}\right\}$$

In addition to considering the costs of false classification, EMP also provides an optimal cut-off point, which is determined as the argument that maximizes the internal expected profit function. However, the downside of this method is that both the value of Return on Investment (ROI) and LGD distribution are assumed to be constant and need to be arbitrarily defined. In reality, these values can change in time and vary depending on specific loan characteristics. Therefore, it is a common practice to create models predicting LGD for each credit separately [29,30]. Additionally, the assumption of a particular form of LGD distribution creates the possibility for a systematic estimation error when this condition is not met.

### 2.4. Calculated Profit

Effective use of the most common metrics comes with limitations, such as the need to work with a balanced set and equality of classification error costs, and to reconcile the representation of the model’s effectiveness only for a particular cut-off point. Even using more advanced metrics tailored to a specific problem means coming to terms with some simplification, as shown for the EMP.

To eliminate problems associated with the use of synthetic metrics as described above, a novel, strictly business-oriented metric called Calculated Profit (CP) is proposed. Calculated Profit may be defined as: (9)$$Calculated\;Profit=MAX\left(\sum_{i=1}^{sample\;size}\left(Credit\;Profit/Loss_{i}|score_{i}<t\right)\right)$$
where Credit Profit/Loss${}_{i}$ = empirical profit or loss on a single credit, score${}_{i}$ = predictor’s score for a specific credit and t = cut-off point.

Determining the CP metric requires computing the cut-off point while maximizing its value; therefore, the optimal value of this threshold is simultaneously supplied by the metric:(10)$$t_{optimal}=ARGMAX\left(\sum_{i=1}^{sample\;size}\left(Credit\;Profit/Loss_{i}|score_{i}<t\right)\right)$$

CP is calculated as a sum of profit (or loss) for every credit admitted to the portfolio. A metric constructed this way has many advantages. To start with, it is insensitive to the class imbalance problem, which is connected with the use of basic metrics such as accuracy. Secondly, it incorporates possible differences in misclassification cost of FP and FN predictions. However, compared to the EMP, it has two significant advantages in this field. While calculating the CP, we make no assumptions about the ROI or shape of the LGD distribution; thus, it is resilient to a constant bias or incorrect determination of those values. It also allows the misclassification costs to be time-variant and correlated with features of the credit itself. Therefore, CP should be more efficient and robust, while simultaneously being easier to apply, as it does not require any predetermined parameters of the dataset. Finally, it represents the direct objective of the company. Thus, the model optimized with its utilization should be most closely aligned with the business problem. At a theoretical level, CP is both simpler and more flexible than presented synthetic metrics. One inconvenience of using CP is the need to possess information about the actual profit on a loan. However, this issue is not a problem in real-world applications. In other cases, it can be estimated, for example, based on the reference interest rate.

## 3. Metric Comparison

The purpose of the next stage of the paper is to compare the CP with other metrics used in credit risk modeling. The comparison will be made by developing separate scoring models, which are then compared in terms of total portfolio profitability on a validation set. Apart from the CP, two more previously described metrics, namely, AUC and EMP, differing in the degree of fit to the problem under study, were selected for the comparison.

### 3.1. Methodology

In order to compare model development strategies based on different metrics, two algorithms, widely used in actual business applications, were utilized. The first of them was the Logistic Regression (LR) with a Weight of Evidence (WoE) variable transformation [31]. Logistic Regression is a classical generalized linear method for binary classification. Despite recent advancements in the area of machine learning, it is still prevalent because of its stability on smaller samples, low likelihood of overfitting and explainability [32,33]. The latter is especially important, due to regulations imposed on the financial sector by international institutions [34] which constrain crediting companies to the explicit use of fully explainable models for credit scoring, making the Logistic Regression still a widely used tool. Additional methods, such as WoE transformation, diminish the limitations of linear models and create performance predictive tools comparable with those prepared using modern machine learning algorithms [35]. Weight of Evidence was specially chosen for this study, as it is a method that originated from and is still widely used in preparation of credit risk models [36].

For the second approach, an XGBoost [37] was chosen as a representant of a modern, state-of-the-art class of machine learning algorithms. Although being a relatively new solution, XGBoost has already been implemented in all kinds of applications, including the financial industry in the less restricted areas [38,39]. With the usage of those two algorithms, the models were optimized using three different metrics:AUC is the most universal and basic metric of those presented. For the model optimized by this method, the cut-off point was selected by minimizing the KROC value (Figure 3), which is the distance from the ROC curve to the point (0,1) of its graph [25]. KROC favorites intermediate solutions with a similar level of FP and FN errors when it comes to selecting the cut-off point. Thus, this approach equalizes both misclassification costs and was chosen for the study as a popular reference solution.EMP, using an inbuilt optimal cut-off point recommendation, represents an advanced, problem-dedicated but synthetic solution. The scoring function parameters were estimated as follows:
(11)$$ROI\left(T_{0}\right)=AVG\left(\frac{Credit\;Profit(\;T_{0}|\;good\;credit\;)}{UPB(T_{0})}\right)$$
where $T_{0}$ = the moment of loan acceptance for a portfolio, and UPB = Credit Unpaid Principal Balance.$p_{0}$ and $p_{1}$ were calculated as proportions of credits with LGD values of below 10% or above 90%.
(12)$$LGD=\frac{Credit\;Loss(\;T_{F}|\;bad\;credit\;)}{UPB(T_{0})}$$
(13)$$p_{0}=F_{LGD}\left(0.1\right)$$
(14)$$p_{1}=1-F_{LGD}\left(0.9\right)$$
where $T_{F}$ = the moment of a loan foreclosure and $F_{LGD}$ = LGD empirical distributor.CP is a novel, strictly goal-oriented metric, described in-depth in Section 2.4.

All models went through an optimization process of feature selection, hyperparameter tunning (for XGBoost) and cut-off point calculation, using the subsequent metrics. The feature selection stage was conducted using the permutated feature importance method [40]. Features that improve the model up to the fifth significant digit (according to the relevant metric) were chosen for the final model. In the case of XGBoost, the optimal hyperparameters were determined using 100 iterations of random search. The entire optimization process was performed using 3-fold cross-validation. Considering two alternative default flags, which will be introduced in the next chapter, 12 models were compared at this study stage.

### 3.2. Dataset

#### 3.2.1. Fannie Mae Multifamily Loan Performance Data

Data used in the study comes from an American agency called Federal National Mortgage Association (FNMA), commonly known as Fannie Mae. It is a private, publicly-traded company working in cooperation with the USA government (Government Sponsored Enterprise). By special law, it is authorized to provide credits and credit guarantees. The corporation is not directly financed by the government, although it operates on preferential terms compared to other enterprises of this type. Fannie Mae’s primary focus is to provide liquidity on the secondary mortgage market by securitizing mortgage loans [41].

The database [42] describes 54,160 credits from a “multifamily” category which includes loans granted to enterprises and secured by multi-apartment buildings. Reported monthly data consist of credits acquired between 1 January 2000 and 31 December 2020, representing about 85% of all multifamily credits acquired by Fannie Mae in this period. The file contains 3,607,268 rows and 57 columns. On average, one credit is present in the dataset for about 67 months. By the thematic criterion, we divided features into: identifying credit/transaction (4), loan terms at origination (11), customer repayment capabilities at origination (2), collateral property (9), loan parameters at the acquisition (8), current loan information (8), loan repayment history (1) and describing credit events (21).

The raw dataset was put through a standard process of data preparation. Firstly, credits were removed for which information was available for less than 12 months as not representative. Then, the average interbank lending rate [43] was added as a reference credit rate. On its basis and through a transformation of the original variables, 17 additional predictors were added. Auxiliary features irrelevant to predictions (i.e., credit id) were removed. A full feature list after this step can be found in Table A1 in the Appendix A. Data gaps were marked as a separate category when their source was known or when they represented a significant proportion of data (over 10%). For the rest of the instances, a K-Nearest Neighbors (KNN) imputing with parameter k=5 was applied. With a view of using a Logistic Regression algorithm, categorical variables were binarized. In the end, highly collinear features were removed based on Variance Inflation Factor (VIF) statistic of over 100.

#### 3.2.2. Profit Calculation

For the purpose of using the CP metric, it is essential to obtain information about the profitability of each finalized credit. It is usually not a problem in a business application. However, publicly available databases usually do not share this kind of insight. In the case of the dataset used in the study, the information about a profit was estimated by subtracting capital costs (interbank reference rate) from installments, calculated based on a current unpaid balance of the credit and a credit rate. Both credit rate and the default cost were available for in the unprocessed database as separate variables. All cash flows were discounted based on annual inflation in the USA [44] at the date of observation. Furthermore, administrative costs were considered in the form of 0.1% of Unpaid Principal Balance (UPB) as the credit portfolio admission cost, and 5% of UPB when the loan entered default. Final credit profit was calculated as follows:When credit was matured:
(15)$$Profit\left(T_{0}\right)=\sum_{t=T_{0}}^{T_{M}}\frac{UPB\left(t\right)*\left[r_{C}\left(t\right)-r_{R}\left(t\right)-5\%*1_{Default\;Entrance}\right]}{\prod_{k=T_{0}}^{t}\left[1+\pi \left(k\right)\right]}-0.1\%*UPB\left(T_{0}\right)$$When credit was foreclosed:
(16)$$Profit\left(T_{0}\right)=\frac{Net\;Lifetime\;Loss(T_{F})}{\prod_{k=T_{0}}^{T_{F}}\left[1+\pi \left(k\right)\right]}$$
where UPB(t) = credit Unpaid Principal Balance at the moment of t, $T_{M}$ = credit maturity moment, $r_{C}\left(t\right)$ = credit rate at the moment of t, $r_{R}\left(t\right)$ = reference rate at the moment of t and π(k) = inflation at the moment of k.

#### 3.2.3. Dependent Variable—Default Flag

In the next step, the default flag was prepared as the dependent variable. In order to test the different approaches, the entire study was conducted using two scenarios:60M flag: Dependent variable with a value of “1” when the credit will fall into a delinquency of over 90 days within the next 60 months. Such a flag follows a common definition of the default flag in the financial industry, proposed by the Basel II regulations [34]. Although the standard observation period is 12 months, a longer 60 months window was chosen in this study because of the long-term character of modeled instruments and to increase the observed default ratio.Lifetime flag: dependent variable with a value of “1” when information about the credit foreclosure is available in the whole credit history.

Based on a default flag, the datasets were then separated. For both target variables, observations with unknown target variable were removed, which meant that either the credit repayment had not yet ended or its lifespan was shorter than 60 months in the case of the 60M flag. Next, for the lifetime flag, only the first record of the credit was chosen for the final dataset because of its lifecycle approach. In contrast, in the case of the 60M flag, one record for every 60 months was selected for the modeling dataset. Such treatment allowed for achieving larger training and testing samples while simultaneously avoiding an overlapping observation window. Model overfitting was not expected to be a problem, as the average number of repetitions of a unique credit in the studied set is around 2. Finally, credits delinquent at the time of observation were excluded, and both datasets were divided into training and testing subsamples in 70%/30% proportions. To avoid the time interference and unequal case distribution, samples were drawn in a shuffled and stratified way with respect to the dependent variable. The main characteristics of modeling-ready datasets are presented in Table 3.

An interesting note from Table 3 is that for the 60M flag, the average loss on a bad client is relatively small, almost zero. The source of this phenomenon is explained partially by the credits profit histograms (Figure 4).

As one can see, for the 60M flag, many loans designated as “defaults” will still make a positive profit. The source of this situation may be due to the not entirely commercial nature of Fannie Mae. Loans entering default, for example, may get favorable debt refinancing terms, and thus a high cure rate is achieved for the entire base. In the case of the lifetime flag, the average loss on a bad customer is still lower than the average gain on a good customer, but this time, the disproportion is much smaller. In addition, there is a sharper separation between profitable and lossy customers for the lifetime flag.

### 3.3. Results

The following section is devoted to discussing the results of models trained according to the methodology presented above. All approaches were tested on the same samples. The parameters of the models were chosen in 3-fold cross-validation, and the results refer to models trained on the entire training subset and validated on the test subset. The main calculations were conducted using Python with sklearn, statsmodels and xgboost packages. A dedicated package [45] in R was used to examine the EMP value.

Analyzing the results of the individual models (Figure 5), we can observe, not surprisingly, that both methods incorporating different misclassification costs for FP and FN predictions produced a significantly better results in terms of profit than the general approach using AUC. In each study, the CP metric achieved the highest profit. The largest difference was approximately 4%, compared to the EMP model, in the case of the lifetime default flag and Logistic Regression. This number may seem small, but a 4% gain achieved by using the more tailored model development metric would add significant value to a company’s overall financial account in an actual business application.

As can be seen in Table 4, the most prominent feature of the models developed by CP is the significantly higher cut-off point selection compared to other metrics, subsequently allowing the largest number of loans to be included into the portfolio in each of the study variants examined. An in-depth analysis of the level of the cut-off point and the average profit per loan indicates that the advantage of the CP metric lies not only in the more appropriate choice of the cut-off point. In the case of the lifetime flag and the LR model, we observed that despite including more loans into the portfolio, the average profit per single loan was higher than for the EMP. This example illustrates the point of using goal-oriented metric for the entire model development process, as in this case, the CP-based model was able to catch the differentiating characteristics of the profitable loans. Additionally, the CP metric, in all cases, chose a significantly smaller number of features, which is usually a beneficial factor in terms of the stability and computational complexity of a model. Moreover, the maximum difference in performance between two algorithms was 6.56% in favor of Logistic Regression in the case of the 60M flag and AUC metric, and that was far smaller than the maximum difference between metrics, which amounted to 21.3% in favor of CP over AUC for XGB and also the 60M flag. That example illustrates that the correct selection of model development metric might be much more beneficial than the choice of predictive algorithm. It is also worth noting that Logistic Regression models boasted equal or better total profit results, compared to XGBoost, for the best models for a given default flag.

The obtained results in terms of the comparison between EMP and AUC are in line with those presented in previous studies [7]. Additionally, in that case, EMP was characterized by a higher number of accepted credits and a higher total profit at the price of a lower average profit per individual customer, in relation to AUC.

As observed above, the difference in performance between the CP and EMP metrics was notable for the lifetime flag but only slight in the case of the 60M flag. The source of this phenomenon was that the EMP, as described in the previous chapter, makes specific assumptions about the LGD distribution when estimating the cost of a false positive classification.

The 60M flag is generally an unusual case for which a great proportion of bad clients are still profitable. The EMP does not incorporate this in the estimated loss on the defaulted client. As the LGD is assumed to take values from 0 to 1, EMP has to utilize its transformed distribution, as in Figure 6 on the right. However, in the case under examination, thanks to a specific appearance of the transformed LGD, EMP more accurately approximates the cost of a bad customer as close to zero. Nevertheless, it is still overestimated.

In the case of the lifetime flag, the LGD distribution (Figure 7) behaves closer to what the EMP assumes with all values in the range [0,1]. However, the estimation error is larger because, based on marginal LGD centiles, the expected value of LGD is underestimated by the EMP.

In order to ensure that the results obtained do not depend on the specific characteristics of the dataset under study, an analysis of the stability of the results is presented in the next section.

## 4. Stability Analysis

The following section is devoted to analyzing the sensitivity of solutions as a function of test conditions and dataset characteristics. The study addresses two issues. First, the effects of the level of additional administrative costs on the performances of individual metrics are tested, as these values were taken arbitrarily and were not directly derived from empirical data. Next, the impacts of the LGD distribution in the study sample on the EMP and CP metrics performance are examined.

### 4.1. Different Levels of Service Costs

The values of administrative costs were assumed in the baseline study to be 0.1% and 5% of current UPB, respectively, for every granted loan and for serving a delinquent credit, though these can vary significantly depending on the nature of the lending institution’s business. These values may be lower for banks working with large, multi-year loans. Simultaneously, they will be much higher in for small, short-term cash loans. That is the reason why this chapter is dedicated to testing the sensitivity of the results as a function of the level of additional costs.

#### 4.1.1. Low Costs

First, a check was conducted for the lower level of costs, where the expense of taking on the portfolio was zero, and the cost of servicing the loan in delinquency amounted to 1% of UPB.

The reduced cost level did not significantly affect the profit on good customers for both default flags (Table 5). However, there were interesting changes on the data side for customers marked as bad, as for the 60M flag, on average, a profit of $118,050 was made on them, and for the lifetime flag, the average loss dropped by about 10%.

In spite of that, the low-cost variant did not noticeably change the visual profit distributions for “good” and “bad” clients, in either the 60M or the lifetime flag case (Figure 8).

Reducing the level of additional costs did not dramatically affect the results of individual metrics, but the changes are noticeable (Figure 9). First, there were slight decreases in the advantages of the CP and EMP over AUC for the lifetime flag. For the 60M case, profit gains on the XGBoost model dropped significantly, by about 10%, for both the EMP and CP metrics. An interesting result of this change is that for both flag variants, the most significant gains from using the problem-tailored metric were achieved by models based on LR. Despite the changes, the models developed with CP again resulted in the most profitable portfolios for each study variant.

A more profound analysis (Table 6) of the details provided similar conclusions as in the case of the baseline study. Again, the CP metric selected the smallest number of variables on average and allowed the largest number of credits per portfolio. Again, the maximum profit difference between two algorithms for the same metric and data subset (6.23% for Lifetime flag, AUC) was much smaller than the difference between metrics with regard to the same algorithm and subset (15.95% for LR, 60M Flag).

#### 4.1.2. High Costs

Next, the models were compared when the cost of admission was 10%, and the expense of servicing the delay amounted to 20% of UPB.

This time the cost alteration had a bigger impact on both subsamples (Table 7). The average profits on good clients decreased by 29.9% and 36.3%, respectively, for the 60M and lifetime flags. Additionally, the loss on bad clients increased drastically for the 60M flag by 1699% to 1.45M USD, almost equalizing the profit. For the lifetime flag, that increase amounted to 72%.

In Figure 10 we observe that the profit histograms in both cases shifted noticeably to the left, and the loss distributions on defaults were stretched. For the lifetime flag, the profit on a portion of “good” customers took a negative turn.

The advantage of the EMP and CP metrics over the AUC also declined in this study variant (Figure 11). This is particularly evident for the models based on the XGBoost algorithm and in the case of the lifetime subbase for the Logistic Regression model developed with EMP. Again, models based on the LR algorithm achieved greater gains for both flags. As in previous studies, CP-based models produced the largest gains for each scenario tested.

On close inspection (Table 8), a pattern familiar from previous studies is repeated. The CP metric selects the lowest number of variables and grants the most credits. However, it is interesting to note that in the case of the XGBoost algorithm, CP selected a lower absolute cut-off point level than EMP. This may indicate a better fit of the model, developed through CP, to the problem under study. A disturbing sign may be that for each case, CP recorded the lowest level of average profit per credit, which was a result of granting the largest number of credits while allowing those with mildly worse parameters also. However, the loss to AUC, which on average boosted the portfolio with the highest average return per loan, did not exceed 3.5% in any case. Once again, the maximum profit difference between two algorithms for the same metric and data subset (4.89% for lifetime flag, AUC) was significantly smaller than the difference between metrics with regard to the same algorithm and subset (10.92% for LR, 60M flag). Even if we compare CP with EMP, then the difference for lifetime flag and LR is still bigger (5.69%) than the maximum difference between the two algorithms.

#### 4.1.3. Cost Comparison Summary

Regardless of the level of costs, which influenced the distribution of the profit on the credits, a similar pattern of results was observed: the CP metric outperformed the EMP and AUC in terms of the total profit, and the difference between metrics was bigger than the gap between algorithms. However, compared to the baseline study, both problem-suited methods provided lower gains over the AUC in terms of total profit than the XGBoost-based models. On average, CP chose the smallest number of features and granted credits to the highest number of borrowers, accepting a slightly lower average profit per customer. It is worth mentioning that those results were very robust and repeated over all variants of the cost levels. Finally, there was no clear dependence between the level of costs and the relative performance of the EMP and CP metrics.

### 4.2. LGD Distributions

As shown above, a major drawback of the EMP metric is the assumption of a specific, point-continuous LGD distribution, which limits the applicability of this metric to the specific cases when this assumption is met. Additionally, EMP assumes a constant ROI and expected LGD value for all the loans in sample. In reality, both ROI and LGD can be dependent on the characteristics of the loan itself, which, if caught by the model, can yield additional gains over the portfolio. Here, the advantage of the CP metric over the EMP metric is that it does not make any assumptions about the distributions of gains and losses, which gives the model additional degrees of freedom to adjust for characteristics of profitable or lossy loans. This section explores how much deviation from the assumptions of specific LGD distribution affects the EMP performance.

#### 4.2.1. Methodology

In order to demonstrate CP’s advantage in the described field, a study on a bootstrapped sample of credits with a pre-defined LGD distribution was conducted. Bootstrapping was used for two purposes. First, the base set is specific, having a very low default ratio and a relatively small cost of granting credit to a bad borrower. This fact directly favors models that admit more loans into the portfolio. To present a more general case, the modeling sample was randomized to achieve a default ratio of 20%. Secondly, bootstrapping was applied to obtain specific distributions of LGD. Four configurations were tested in the study:Uniform distribution between 0 and 1.EMP-assumed, discrete–continuous probability distribution with discrete weights at the ends of the interval ($p_{0}$ = 0.5 and $p_{1}$ = 0.25); uniform distribution on the interval (0, 1).Right-skewed beta distribution with parameters alpha and beta equal to 3 and 8.Left-skewed beta distribution with parameters alpha and beta equal to 8 and 3.

Example drawn samples are presented in Figure 12.

The study results were averaged from 100 bootstrapped samples. To reduce subsample instability, 10% of loans with the highest level of UPB at the time of acquisition were excluded from the draw. Without this step, single sample profits were largely defined by whether or not the model accepted single, dominating credits with a large UPB, requiring many more draw repetitions to achieve reproducible results for the whole study. A single sample consisted of approximately 200 defaults and 800 non-defaulted credits. Analogously to the baseline study, EMP and CP were considered for variable selection, hyperparameter optimization and cut-off point determination.

#### 4.2.2. Bootstrapping Results

As can be seen in Figure 13, the results of the EMP metric are comparable to those of CP for the uniform and EMP-assumed distribution. Such an outcome was quite expected because, for these examples, the actual expected value of the LGD matches that estimated by the EMP. The CP’s advantage is revealed for both beta distributions. In the case of a right-skewed distribution, EMP overestimates the cost of the default and allows fewer loans in the portfolio than is optimal. For a left-skewed distribution, the empirical LGD is significantly higher than the one determined by the EMP, thereby granting too many loans to lossy customers. As in previous studies, LR-based models repeatedly outperformed XGBoost.

Table 9 shows that EMP distribution parameters ($p_{0}$, $p_{1}$) were correctly estimated for all variants of the test. However, their values did not allow is to properly approximate the properties of the samples for both beta distributions. Once again, the CP metric selected the fewest predictors for the model in all the cases considered.

## 5. Conclusions

We aimed to present a new empirical, business-oriented metric for evaluating predictive credit risk models, Calculated Profit, and to compare it with other metrics used for this purpose (AUC, EMP). The study was based on the “Multifamily” loan database acquired from the Fannie Mae. In the research, Calculated Profit outperformed the problem-tailored metric EMP, by up to 4% in terms of total portfolio profit on the test subset, and the outperformance was even greater compared to the more generic metric AUC. The sensitivity analysis demonstrated that this advantage is stable for different dataset characteristics. The results indicate that applying CP to the entire development process gives the model additional degrees of freedom to better adapt to the nature of the problem, allowing for additional advantages. Furthermore, in the studied case, the differences between the compared metrics proved to be significantly bigger than the differences between the used predictive algorithms (Logistic Regression and XGBoost), highlighting the importance of choosing the right information metric for the model development process. A conclusion of the study is that in real applications, the usage of artificial, even problem-dedicated assessment methods for model development can lead to suboptimal results. This study demonstrated that top-down assumptions negatively affect the performance and overall applicability of model evaluation metrics. The general definition of the CP metric makes it both more efficient and more flexible, making it also suitable for applications other than credit risk.

It is also worth mentioning that credit scoring is a multidimensional process that is evaluated not only through the perspective of direct profits. Financial institutions need to manage the portfolio risk at least equally strongly. In this respect, CP may lead to a riskier result, as in many cases, it allows more credit into the portfolio. However, this issue does not undermine the concept of CP as a profit-driven metric. It would be intriguing to develop this method into a multidimensional approach. In this way, other factors, such as the described risk perspective, could be incorporated into the model development process. In addition, it would also be valuable to expand the study to include a comparison of CP’s performance with those of other metrics, based on other datasets and using other predictive algorithms.

## Figures and Tables

**Figure 1 entropy-24-01218-f001:**
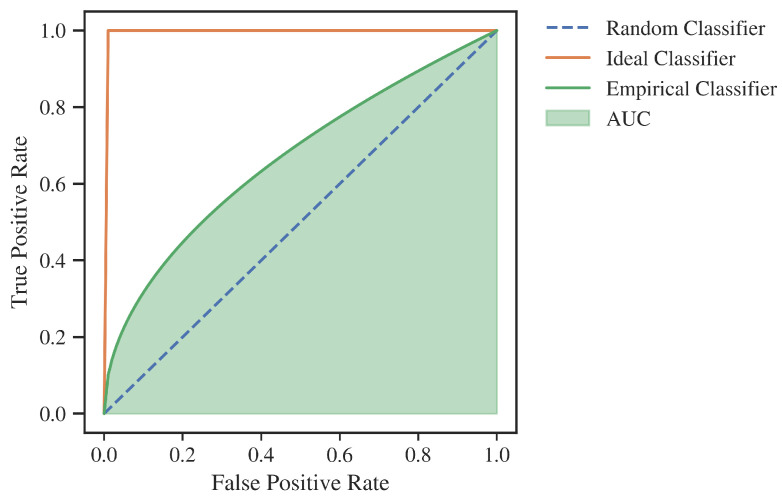
Area Under the ROC Curve—example. Source: own preparation.

**Figure 2 entropy-24-01218-f002:**
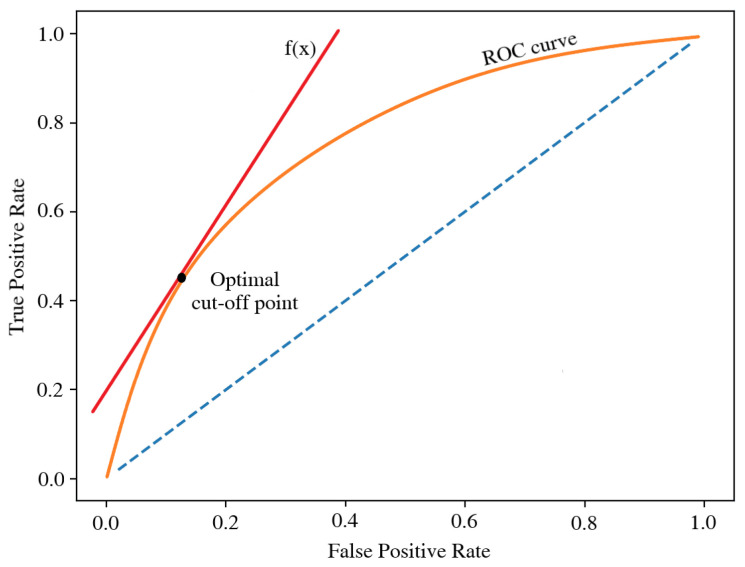
ROC-based, cost-sensitive method for determining the optimal cutoff point. Source: own preparation.

**Figure 3 entropy-24-01218-f003:**
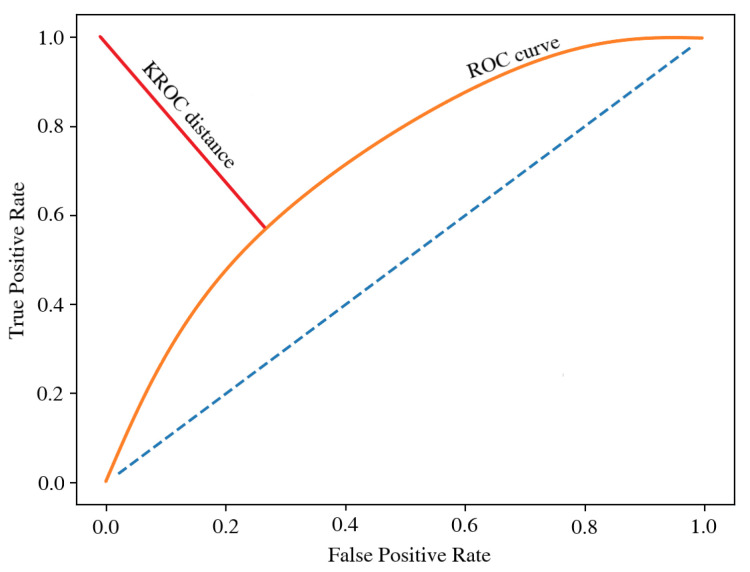
KROC value. Source: own preparation.

**Figure 4 entropy-24-01218-f004:**
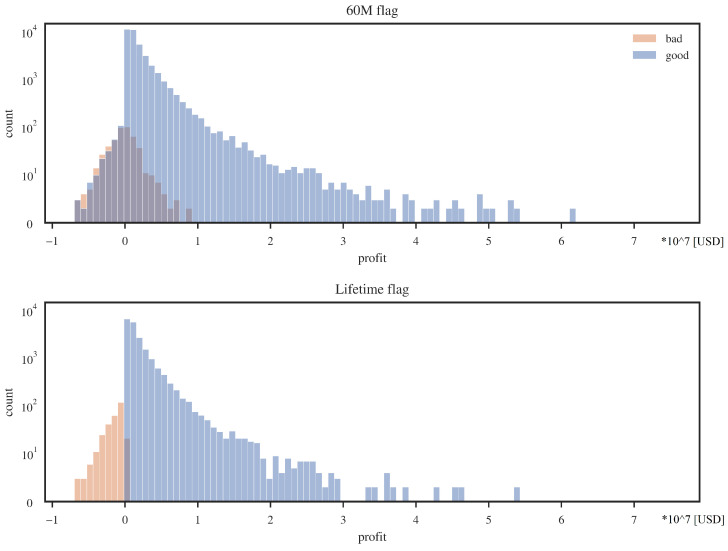
Profit distributions for “good” and “bad” credits; two datasets for alternative default flags. Source: own preparation.

**Figure 5 entropy-24-01218-f005:**
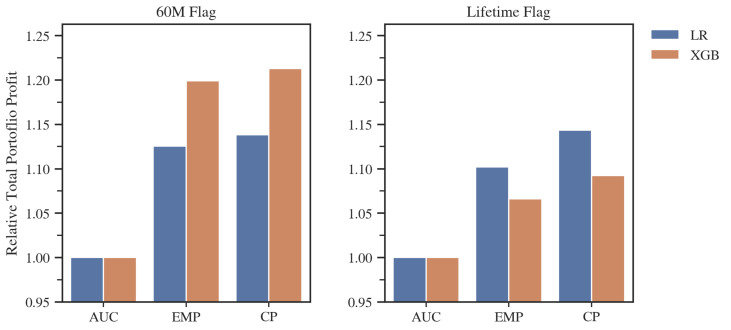
Total portfolio profit on the validation set for models developed with different metrics, standardized to the result of the AUC-developed models. Source: own preparation.

**Figure 6 entropy-24-01218-f006:**
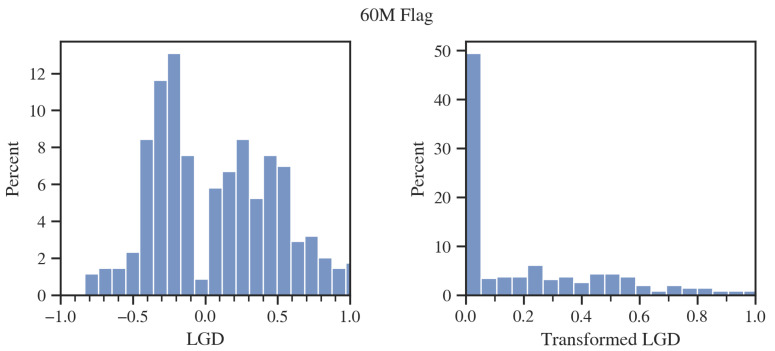
LGD histogram for the 60M flag. Source: own preparation.

**Figure 7 entropy-24-01218-f007:**
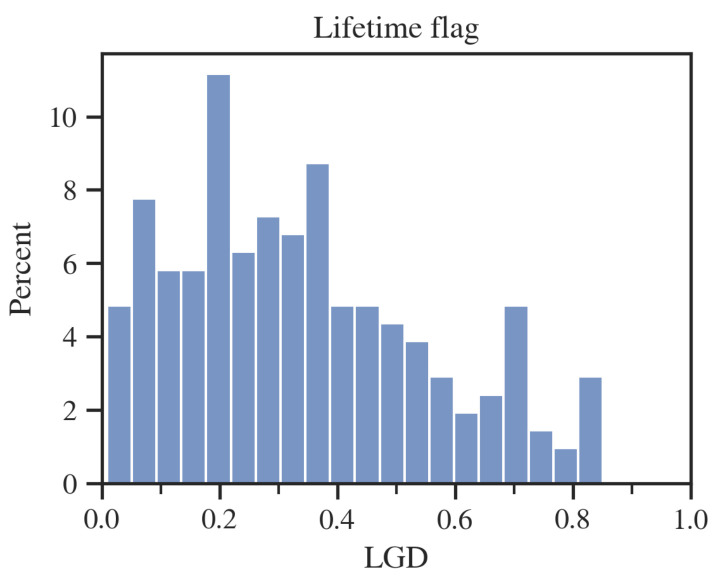
LGD histograms for the lifetime flag. Source: own preparation.

**Figure 8 entropy-24-01218-f008:**
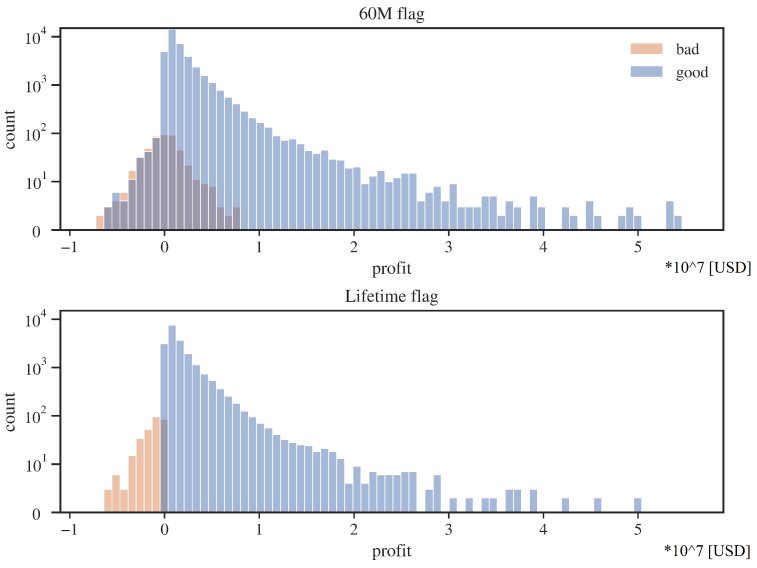
Profit distributions for “good” and “bad” credits, low administrative costs variant. Source: own preparation.

**Figure 9 entropy-24-01218-f009:**
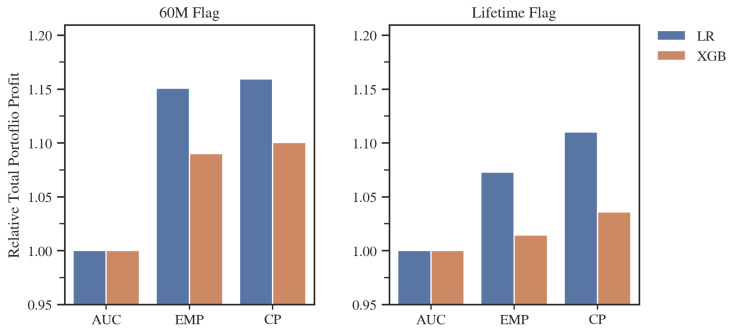
Total portfolio profits on the validation set for models developed with different metrics, standardized to the result of the AUC-developed models, low administrative costs variant. Source: own preparation.

**Figure 10 entropy-24-01218-f010:**
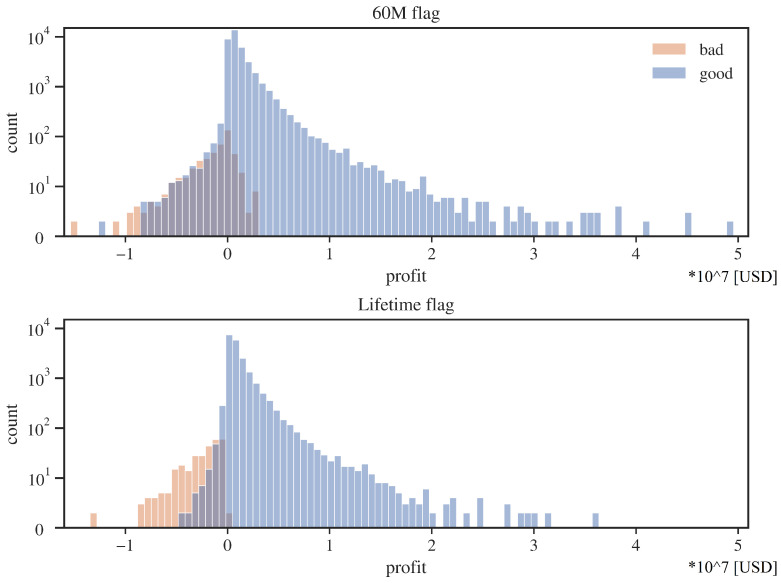
Profit distributions for “good” and “bad” credits, high costs. Source: own preparation.

**Figure 11 entropy-24-01218-f011:**
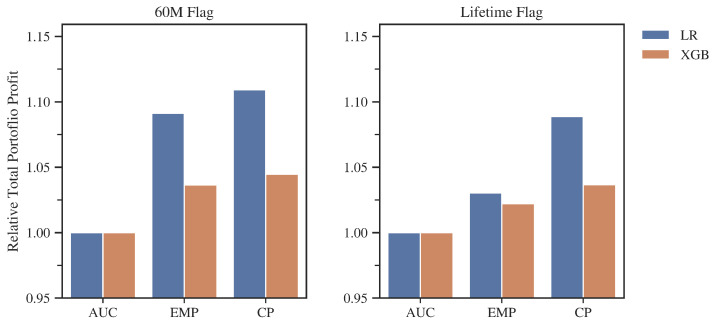
Total portfolio profit on the validation set for models developed with different metrics, standardized to the result of the AUC-developed models, high administrative costs variant. Source: own preparation.

**Figure 12 entropy-24-01218-f012:**
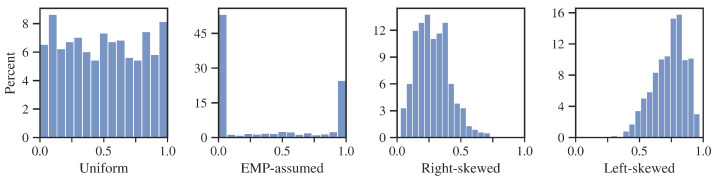
Example LGD distributions from a single drawn sample. Source: own preparation.

**Figure 13 entropy-24-01218-f013:**
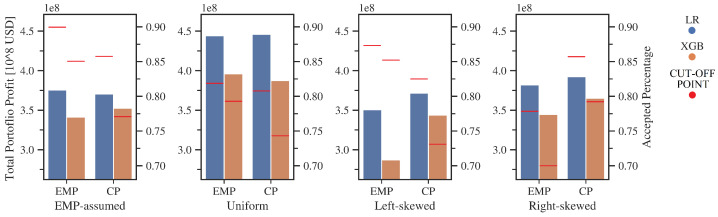
Averaged metrics results for bootstrapped samples. Total portfolio profit on the left scale. Percentage cut-off marked on the right. Source: own preparation.

**Table 1 entropy-24-01218-t001:** Contingency table—possible binary classification results; false classifications marked in gray. Source: own preparation.

		Prediction	
		Bad	Good
**Reality**	Bad	True Positive (TP)	False Negative (FN)
	Good	False Positive (FP)	True Negative (TN)

**Table 2 entropy-24-01218-t002:** Model payoff table for binary classification. Source: own preparation.

		Prediction	
		Bad	Good
**Reality**	Bad	Denying credit to bad client cost = 0	Granting credit to bad client cost = loss on default
	Good	Denying credit to good client cost = loss of profit on good credit	Granting credit to good client cost = 0

**Table 3 entropy-24-01218-t003:** Datasets’ characteristics. Source: own preparation.

Default Flag	Number of Records	Unique Credits	Number of Features	Default Ratio	Average Profit on a “Good” Client [USD]	Average Profit on a “Bad” Client [USD]
60M	39,086	20,354	94	1.26%	2,213,795	−80,495
Lifetime	20,370	20,370	94	1.44%	2,054,782	−1,498,701

**Table 4 entropy-24-01218-t004:** LR and XGB models results for 60M and lifetime flags—baseline study. Source: own calculations.

Flag	Model	Metric	Number of Features	Cut-off Point ^1^	Number of Granted Credits	Average Profit on Credit $[*10^{6}\;$ USD]	Total Portfolio Profit $[*10^{10}\;$ USD]
60M	LR	AUC	64	0.015	10,303	2.176	2.242
		EMP	53	0.104	11,587	2.178	2.523
		CP	15	0.406	11,721	2.177	2.552
	XGB	AUC	37	0.419	9199	2.287	2.104
		EMP	44	0.744	11,515	2.191	2.523
		CP	28	0.546	11,726	2.176	2.552
Lifetime	LR	AUC	64	0.013	5328	1.989	1.060
		EMP	69	0.07	5931	1.970	1.168
		CP	13	0.754	6111	1.983	1.212
	XGB	AUC	41	0.521	5251	2.108	1.107
		EMP	25	0.609	5873	2.009	1.180
		CP	23	0.877	6076	1.990	1.209

^1^ A relatively high cut-off point for the XGB-based models arises from a correction with an in-built parameter for
imbalanced samples.

**Table 5 entropy-24-01218-t005:** Datasets’ characteristics, low administrative costs variant. Source: own calculations.

Default Flag	Average Profit on a “Good” Client [USD]	Average Profit on a “Bad” Client [USD]
60M	2,223,944	118,050
Lifetime	2,064,946	−1,347,192

**Table 6 entropy-24-01218-t006:** LR and XGB models’ results for 60M and lifetime flags—low administrative cost variant. Source: own calculations.

Flag	Model	Metric	Number of Features	Cut-off Point	Number of Granted Credits	Average Profit on Credit $[*10^{6}\;$ USD]	Total Portfolio Profit $[*10^{10}\;$ USD]
60M	LR	AUC	65	0.015	10,201	2.169	2.213
		EMP	52	0.104	11,593	2.197	2.547
		CP	18	0.566	11,723	2.189	2.566
	XGB	AUC	42	0.464	10,212	2.281	2.330
		EMP	37	0.699	11,505	2.208	2.540
		CP	33	0.673	11,713	2.189	2.564
Lifetime	LR	AUC	65	0.017	5413	2.031	1.099
		EMP	65	0.081	5939	1.986	1.179
		CP	14	0.861	6111	1.996	1.220
	XGB	AUC	35	0.540	5728	2.046	1.172
		EMP	26	0.746	5875	2.023	1.189
		CP	30	0.908	6079	1.997	1.214

**Table 7 entropy-24-01218-t007:** Datasets’ characteristics, high administrative costs variant. Source: own calculations.

Default Flag	Average Profit on a “Good” Client [USD]	Average Profit on a “Bad” Client [USD]
60M	1,452,320	−1,447,737
Lifetime	1,312,816	−2,578,391

**Table 8 entropy-24-01218-t008:** LR & XGB models results for 60M and Lifetime flags - high administrative costs variant. Source: Own calculations.

Flag	Model	Metric	Number of Features	Cut-off Point	Number of Granted Credits	Average Profit on Credit $[*10^{6}\;$ USD]	Total Portfolio Profit $[*10^{10}\;$ USD]
60M	LR	AUC	59	0.014	10,376	1.448	1.502
		EMP	83	0.053	11,426	1.434	1.639
		CP	14	0.561	11,724	1.421	1.666
	XGB	AUC	50	0.461	11,004	1.446	1.591
		EMP	42	0.668	11,457	1.439	1.649
		CP	39	0.572	11,638	1.428	1.662
Lifetime	LR	AUC	55	0.02	5510	1.269	0.700
		EMP	55	0.042	5756	1.252	0.721
		CP	14	0.462	6101	1.248	0.762
	XGB	AUC	45	0.595	5650	1.302	0.736
		EMP	32	0.594	5901	1.274	0.752
		CP	23	0.534	6063	1.258	0.763

**Table 9 entropy-24-01218-t009:** Detailed averaged metrics results for bootstrapped samples, Source: Own calculations.

Profit	Model	Metric	$p_{0}$ ^1^	$p_{1}$ ^1^	Number of Features	Accepted Percentage	Total Portfolio Profit [$*10^{8}$ USD]
EMP -assumed	LR	EMP	0.524	0.276	56.9	89.8%	3.75
		CP			23.7	86.0%	3.71
	XGB	EMP			25.9	85.3%	3.41
		CP			20.8	77.1%	3.52
Uniform	LR	EMP	0.094	0.105	50.0	82.1%	4.44
		CP			47.6	81.4%	4.46
	XGB	EMP			37.7	79.1%	3.96
		CP			21.1	74.3%	3.87
Beta right	LR	EMP	0.068	0	38.1	78.3%	3.82
		CP			18.9	86.5%	3.92
	XGB	EMP			23.9	70.0%	3.45
		CP			22.8	79.1%	3.65
Beta left	LR	EMP	0	0.072	60.9	87.3%	3.51
		CP			36.8	82.5%	3.71
	XGB	EMP			32.9	85.5%	2.87
		CP			32.8	73.3%	3.44

^1^*p*_0_, *p*_1_ are the calculated EMP coefficients for the estimated LGD distribution.

## Data Availability

Dataset used in the study was derived from: https://capitalmarkets.fanniemae.com/credit-risk-transfer/multifamily-credit-risk-transfer/multifamily-loan-performance-data (accessed on 18 March 2021).

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
