# Peer review of "The Advantage of Case-Tailored Information Metrics for the Development of Predictive Models, Calculated Profit in Credit Scoring"

_entropy, 2022, doi:10.3390/e24091218_

Round 1
Reviewer 1 Report
Title: The importance of using business-oriented metrics for the development of predictive models, application of Calculated Profit and EMP metrics in credit scoring
This paper addresses an interesting and timely issue in building a credit scoring model: using a business-oriented metric for developing a predictive model. This paper proposes a case-tailored calculated profit (CP) as a novel business-oriented metri, and compares the logistic regression and XGboost machine learning algorithms with two existing metrics: the Areal Under the ROC curve (AUC) and problem dedicated expected maximum profit (EMP).
My concerns are described as follows. Indeed, it is already known that different metrics result in different thresholds in building credit scoring models. Thus, the contribution of this paper appears to be marginal to the literature. In addition, because this paper focuses on proposing a new metric, it is suggested to highlight the benefit of the proposed CP in more details in terms of theoretical, numerical, or financial aspects. A more complete survey on existing metrics in the literature can be improved as well.
References
A. E. Khandani, A. J. Kim, and A. W. Lo, “Consumer credit-risk models via
machine-learning algorithms,” Journal of Banking and Finance, vol. 34, no. 11, pp. 2767 – 2787, 2010. [Online]. Available: http://www.sciencedirect.com/science/article/pii/
S0378426610002372
Author Response
Authors’ response:
- The aim of the paper was to present a new business-oriented method for assessing credit risk models and to compare it with other metrics used for this purpose, based on empirical data. Demonstrating that different metrics result in the selection of a different cut-off level was part of the analysis of the results, indirectly guiding but not defining the final performance of each metric. The analysis of threshold selection was therefore not the main aim of the paper. There is evidence in the paper that the advantage of the CP metric is not limited to the selection of a more accurate cut-off point. Regarding the contribution to the current state of the literature, we would like to point out that the paper: Firstly, highlights the importance of selecting the right assessment metric used for model development and, on empirical data, presents the advantage that can be achieved by using a more appropriate one. Secondly, it presents a new metric (CP), tailored to the problem of credit scoring that can provide such an advantage. Thirdly, it shows that the selection of metric used for model development can be more important than the choice of the predictive algorithm itself. In our view, all three of these areas are so far poorly explored and described when it comes to building predictive models for finance applications. Addressing the comments sent, the article has been rewritten to present these aspects more directly.
- The article has been rewritten to more directly present the aims of the research and to highlight the advantages of the Calculated Profit metric.
- The literature overview has been revised and expanded.
Reviewer 2 Report
entropy-1844176-peer-review-v1:
The importance of using business-oriented metrics for the development of predictive models, application of Calculated Profit and EMP metrics in credit scoring
Recommendation: Major revision
Summary of this paper:
The paper proposes a new case-tailored metric called Calculated Profit (CP), and contrasts it with generic metrics, such as the area under the receiver operating characteristic curve (AUC), and business-oriented metrics, such as Expected Maximum Profit (EMP). With the multifamily loan data acquired from the Fannie Mae database, the authors set up a profit maximization problem for the loan issuer who needs to decide whether or not to grant credit to a client based on the prediction about the client’s default. The authors model the credit risk scoring by two predictive algorithms, namely Logistic Regression (LR) and extreme gradient boosting machine (XGB), and use different metrics to determine the cut-off point of the binary classifier that predicts default.
This paper provides empirical evidence that the proposed CP metric outperforms EMP and AUC in total profit maximization. Meanwhile, the CP metric selects fewer features for the prediction model. The sensitivity analysis demonstrates that this advantage is stable for different dataset characteristics.
Comments:
1) The contribution of the CP metric
I read this paper with great interest. The idea of a case-tailored metric could be essential for a prediction model. From my own perspective, the main contribution of this paper is that the authors propose a new case-tailored metric called Calculated Profit (CP) and illustrate the advantages of such metric with empirical evidence.
In this sense, the authors should consider highlighting the contribution of the CP metric through some more comprehensive analyses. For instance, I would expect to see more out-of-sample performance comparisons among a wider range of alternative prediction algorithms, feature selection methods, and generic/business-oriented metrics so that the readers would be more convinced that
a) the choice of different metrics matters as important as, or even more important than, the choice of different algorithms and feature engineering methods
b) the newly proposed CP metric significantly outperforms the existing ones in terms of profit maximization and/or model efficiency.
2) Title of the paper
This paper is not the pioneering work[1] studying the advantage of profit-based classification metrics over generic classification metrics. The authors should change their title to reflect the fact that their main contribution is to establish a new profit-based and case-tailored classification metric, namely CP, that outperforms the existing ones.
3) Feature selection for XGB
In the last paragraph in section 3.1, the authors explain in detail how they select features for the LR model but fail to do so for the XGB model.
4) Minor comments
The first time the authors mention an abbreviation or acronym, they should spell it out in full and provide the abbreviation in parentheses. (e.g., UPB in the first paragraph of section 3.2.2.)
[1] See, for example, Thomas Verbraken, Cristia ́n Bravo, Richard Weber, and Bart Baesens. 2014. Development and application of consumer credit scoring models using profit-based classification measures. European Journal of Operational Research, 238:505–513.
Author Response
Authors’ response:
- The article has been rewritten to better present the advantages of the CP metric and to show directly that the right choice of information metric may be more important than the choice of predictive algorithm. The changes involve revision of the theoretical presentation of the CP, analysis of the results and the conclusion. Regarding the question of a more extensive comparative analysis using additional metrics, other datasets and predictive algorithms, we agree that this would be an interesting continuation of the study. However, the aim of the study was to present the new metric, describe its features from the theoretical perspective and validate it on an empirical set using leading algorithms and the most popular reference metrics. We believe that the results obtained in this study confirm the theoretical merits of the CP. Extending the comparison would be valuable, but it would not change the fundamental advantages of CP or the conclusions about the importance of selecting appropriate metrics for the model development process.
- The title has been reformulated based on this and the editor’s comment.
- The last paragraph of section 3.1 has been reformulated to clarify that the optimization framework was similar for both predictive algorithms.
- Acronyms' definitions were corrected.

Reviewer 3 Report
The paper proposes a formula to assess credit risk and describes experiments with Logistic Regression and XGBoost to determine the performance of the formula. In general the experiments are well designed and the paper is well written and easy to follow. Here are some comments to hopefully help improve the paper.
- In the process, a number of features are selected from the dataset. Then, the algorithms are used to also select the best of the remaining features. The paper could clarify which features are selected and why they are deemed important, specially considering that there is a large reduction in the number of features in some experiments. This is also mandatory for repeatability of the study.
- The platform used for the experiments could also be clarified in the paper.
- At line 247 it is mentioned that KNN is used as inputation method for missing data. More details about the process and its impact on the data could be given.
- The confusion matrix presented in 2.2 is very common in assessment of ML algorithms and other experiments. The reference to a paper about text classification is a bit odd.
- In Table 2 it could possibly be "loss of profit on good credit", rather than "profit on good credit"
- At the end of the introduction there is a reference to "the thesis", which should perhaps be "the paper"
- All acronyms should be defined the first time they are used. Some are not defined, or are defined long after the first use.
- The references should be in MDPI format, or at least sorted; the way they are it is difficult to find a reference.
Author Response
Authors’ response:
- Chapter 3.2.1 describes the preliminary feature engineering step. According to information there, the original dataset includes 56 exogenic variables, which were put through the following preparation steps:
- Addition of the reference interbank rate
- Addition of a features transformation
- Removal of features irrelevant to prediction, i.e. credit id (example added to the text)
- Binarization of categorical features
After this step, 94 features (information in Table 3) were passed to the models optimization process. Model optimization process explained in the last paragraph of 3.1. section, for each metric and algorithm separately, includes feature selection using Permutated Feature Importance method (reference added to the text). A large reduction in the number of features in some experiments results from this stage of model optimization. We believe that the provided description allows to reproduce our dataset quite accurately. In addition, a complete list of used variables has been included in the appendix.
2. Information about the calculations platform was added at the beginning of the Results section. However, the study can be carried out in any statistical package with XGBoost implementation, without this affecting the results.
3. Imputation of missing data using the KNN method was not given more attention in the paper, as this step is only a minor part of the data preparation process and has little impact on the results of the study. However, in line with a pertinent comment, in order to ensure full reproducibility of the study, this part of the text has been supplemented with information on the size of the K parameter used in the imputation.
4. Reference removed
5. Modified according to the suggestion.
6. Corrected
7. Acronyms were checked and defined at the first occurrence.
8. The article has been adapted to the MDPI format.

Round 2
Reviewer 1 Report
It is recommended to accept this paper at its present form.
Reviewer 2 Report
I am not sure about the style and the bar of this journal. I review more for top business and economics journals. I have quickly read the revised manuscript. The paper now reads OK but the robustness of the results is not rigorous. The improvement brought by this new metric may be rather marginal. I raised the issue on the choice of model and feature, but the author's response is that the current two models are sufficient.
Reviewer 3 Report
I believe the paper has been greatly improved. It is now clearer and more complete. My main concerns have been addressed.